# Cluster randomised trial of a health system strengthening approach applying person-centred communication for the prevention of female genital mutilation in Guinea, Kenya and Somalia

Mamadou Dioulde Balde,[1] Patrick Muia Ndavi,[2] Vernon Mochache,[3] Anne-Marie Soumah,[1] Tammary Esho,[4] James Munyao King'oo,[5] Jackline Kemboi,[2] Alpha Oumar Sall,[1] Aissatou Diallo,[1] Wisal Ahmed,[6] Karin Stein,[7] Khurshed Nosirov,[3] Soe Soe Thwin,[3] Max Petzold,[8] Muna Abdi Ahmed,[9] Ahmed Diriye,[10] Christina Pallitto  [3]

MDB and PMN are joint first authors.

**Correspondence to**
Dr Christina Pallitto;
pallittoc@who.int

## ABSTRACT

**Introduction** There is limited evidence on effective health systems interventions for preventing female genital mutilation (FGM). This study tested a two-level intervention package at primary care applying person-centred communication (PCC) for FGM prevention.

**Methods** A cluster randomised trial was conducted in 2020–2021 in 180 antenatal care (ANC) clinics in Guinea, Kenya and Somalia. At baseline, all clinics received guidance and materials on FGM prevention and care; at month 3, ANC providers at intervention sites received PCC training. Data were collected from clinic managers, ANC providers and clients at baseline, month 3 and month 6 on primary outcomes, including delivery of PCC counselling, utilisation of level one materials, health facility preparedness for FGM prevention and care services and secondary outcomes related to clients' and providers' knowledge and attitudes. Data were analysed using multilevel and single-level logistic regression models.

**Results** Providers in the intervention arm were more likely to deliver PCC for FGM prevention compared with those in the control arm, including inquiring about clients' FGM status (adjusted OR (AOR): 8.9, 95% CI: 6.9 to 11.5; p<0.001) and FGM-related beliefs (AOR: 9.7, 95% CI: 7.5 to 12.5; p<0.001) and discussing why (AOR: 9.2, 95% CI: 7.1 to 11.9; p<0.001) or how (AOR: 7.7, 95% CI: 6.0 to 9.9; p<0.001) FGM should be prevented. They were more confident in their FGM-related knowledge (AOR: 7.0, 95% CI: 1.5 to 32.3; p=0.012) and communication skills (AOR: 1.8; 95% CI: 1.0 to 3.2; p=0.035). Intervention clients were less supportive of FGM (AOR: 5.4, 95% CI: 2.4 to 12.4; p<0.001) and had lower intentions of having their daughters undergo FGM (AOR: 0.3, 95% CI: 0.1 to 0.7; p=0.004) or seeking medicalised FGM (AOR: 0.2, 95% CI: 0.1 to 0.5; p<0.001) compared with those in the control arm.

**Conclusion** This is the first study to provide evidence of an effective FGM prevention intervention that can be

## STRENGTHS AND LIMITATIONS OF THIS STUDY

⇒ This hybrid-effectiveness implementation research study conducted in primary care public health facilities in three countries with a high prevalence of female genital mutilation (FGM) assessed the role of health workers in providing FGM prevention communication in the context of routine antenatal care (ANC).

⇒ It will be important to assess the effectiveness of the person-centred communication approach in other service delivery points, for example, child immunisation, and with other cadres of health workers, for example, community health workers, to assess its effectiveness beyond ANC.

⇒ Many factors influence FGM-related decision-making, and while primary care health workers were found to be effective communicators, and the randomised design controlled for some external factors, the impact of a health sector intervention in conjunction with multisectoral initiatives requires further investigation.

⇒ To ensure participation of at least one ANC provider at each site through each time point, eligibility of health workers was based on clinical rotation schedules, which may have introduced a selection bias although the included and excluded providers did not appear to differ significantly.

delivered in primary care settings in high-prevalence countries.

**Trial registration and date** PACTR201906696419769 (3 June 2019).

## INTRODUCTION

Multisectoral efforts are needed to achieve Sustainable Development Goal 5.3 to

eliminate the harmful practice of female genital mutilation (FGM) by 2030 in line with the United Nation's (UN) General Assembly resolution 67/146,[1] the World Health Assembly Resolution 61.16[2] and the 2008 Interagency Statement,[3] which call on UN Member States to enact comprehensive and multidisciplinary national action plans and strategies towards the elimination of the practice. Identifying effective strategies across sectors is an important step in ending FGM.

The health system, defined as all organisations, institutions and resources that produce actions whose primary purpose is to improve health,[4] has an important role to play not only in managing complications of FGM but also in preventing the practice. Healthcare providers, specifically nurses and midwives who constitute most of the health workforce, are highly respected members of FGM practicing communities and could positively contribute to abandonment efforts.[5 6] However, there is currently limited evidence to guide health programming on FGM prevention.[7] In addition, some healthcare providers are themselves supportive of this harmful practice, and might even perform it (ie, FGM medicalisation), despite national laws and medical ethics forbidding it.[8–11] Developing evidence-based tools to build the skills of healthcare providers and address their underlying beliefs could contribute to FGM abandonment efforts and complement existing resources on the management of complications[12 13] to ensure comprehensive and high-quality care.

Three countries (Guinea, Somalia and Kenya) participated in a cluster randomised trial to test the effectiveness and implementation of a health system strengthening approach to FGM, which included the testing of an intervention to build skills of health workers on applying person-centred communication (PCC) for the prevention of FGM.[14] Study countries were selected based on their high national and/or subnational FGM prevalence. The national prevalence of FGM among women and girls aged 15–49 years is 98% in Somalia, 97% in Guinea and 21% in Kenya according to national population-based surveys. There are 20 hotspot counties/subnational administrative units in Kenya with a prevalence of >80%,[15] and this study focused on three of these counties. Likewise, the study countries have high rates of medicalised FGM, performed primarily by midwives, who make up between 71% and 93% of primary healthcare providers in the three study countries[16] hence the selection of nurses and midwives as the target group for this intervention.

The purpose of this study was to test a two-level intervention package to enable antenatal care (ANC) providers to deliver person-centred FGM counselling to their clients.[1] This intervention package was informed by a theory of change that promotes health workers to be effective behavioural change agents because of their credibility[17] and positionality to influence the opinions, attitudes, beliefs, motivations and behaviours of their clients.[18] We hypothesised that if ANC providers gained the necessary knowledge and skills to provide person-centred counselling (level two) and were given the opportunity to question their beliefs and attitudes together with an enabling environment (level one), they could positively influence the knowledge and attitudes of their clients to abandon the practice (online supplemental file 1).

The level one intervention consisted of making available national policy directives on the role of healthcare providers in providing FGM prevention and care services, WHO's FGM guidelines and clinical handbook as well as information, education and communication (IEC) materials. These materials were distributed without any capacity building to accompany their distribution. Level two consisted of an interactive training specifically targeting ANC providers to build their knowledge on FGM, enable them to question their FGM-related values and attitudes and build their skills on counselling for FGM prevention using PCC,[19] a component of person-centred care, which ensures that the perspectives and preferences of individuals, carers, families and communities are at the centre of decisions and that they have the information and support needed to make decisions.[20] ANC providers were trained to apply a series of structured steps in which they would: 'Assess' their client's views on FGM, address and challenge her 'Beliefs', encourage 'Change' and together with the client, 'Discuss and Decide' (ABCD).

## METHODS
### Study design
This cluster randomised trial applied a type 2 hybrid, effectiveness-implementation design[21] to test the effectiveness of the delivery of a phased intervention package (level one and two) on knowledge, attitudes and practices among ANC health workers and their clients. This type of implementation research design assesses the effectiveness of the intervention and implementation factors in real-world settings. The methodology, analysis plan and reporting conformed to the Consolidated Standards of Reporting Trial 2010 statement: extension for cluster randomised trials checklist.[22] Each study country submitted country-specific protocols to local institutional review boards.

### Participants
Within each study country, two or three subnational units (regions/counties) were purposively selected according to the following eligibility criteria[1]: FGM prevalence >50% among women 15–49 years old[2]; more than 15 ANC clinics, seeing on average 30 new ANC clients per month and[3] accessibility in terms of security. The unit of randomisation was the ANC clinic to avoid having ANC providers in the same clinic in different study arms, which could lead to contamination. In intervention sites, all providers on duty were pre-screened. To ensure participation and follow-up throughout the trial, between one and three ANC providers on duty were enrolled based on a 6-month clinical rotation schedule provided by the clinic manager. 10 new clients exiting their first ANC consultation with a

participating provider were recruited at each data collection point.

Individual study participants gave verbal informed consent. Data collectors collected data from the ANC providers and their clients in a private and confidential setting. While personally identifiable information was collected from ANC providers to facilitate tracking during the follow-up data collection time points, data were de-identified prior to analysis. No personally identifiable information was collected from ANC clients who were unique at each time point. Participating ANC clients received the equivalent of US$5 to compensate for their transport costs recognising that participants consenting to participate might have changed their plans to accommodate the interviews. Given the insecurity in carrying cash in Somalia, a mobile phone application was used to transfer the money to participants, an amendment to the original protocol, which was submitted to the ethical review committees.

### Randomisation and blinding

Based on Ministry of Health (MoH) facility administrative records from all public, primary care facilities (ie, dispensaries and/or health centres) offering ANC services in the selected regions/counties, the average number of new ANC clients seen in November and December 2019 was compiled to create ordered listings of client loads at each of the sites by region/county. Clinics were matched into pairs based on client load so the two busiest would be randomised to different arms and so on. A uniform distribution was used for randomisation using the uniform random number function in Stata V.17 (StataCorp, College Station, Texas, USA). Study teams organised data collection and intervention trainings based on the randomisation lists. Attempts were made to blind clinic managers, ANC providers and their clients to study arm allocation. Since both study arms received the level one intervention component at baseline, and the providers and managers at control sites were unaware of the training that took place at intervention sites, it is conceivable that they were not aware of their study arm. Presumably, providers at intervention sites would assume they were the intervention arm, but they were also not aware of what might have been offered to other sites. ANC clients, however, were completely blinded as to study arm allocation since a distinct set of clients was interviewed at each time point, and they would not be aware of the training the provider had. Field data collectors were also blinded to study arm allocation as much as possible, although some might have determined the intervention arm during the study.

### Procedures

Implementation of the study interventions and data collection occurred between August 2020 and September 2021 and was staggered by countries. In the intervention arm, data collection was undertaken at three time points, that is, at baseline prior to implementing the level one intervention component; at month 3, prior to

implementing the level two intervention component and at month 6. In the control arm, data collection was done at two time points, that is, at baseline and at month 6. Study instruments included one for ANC clients, one for health workers and a health facility checklist completed by clinic managers. Instruments were pretested among ANC clients and providers from non-participating sites in all countries, and country teams provided feedback on the structure and appropriateness of each question prior to finalising the instruments.

A web-interface electronic data capture system was developed on the Kobo Toolbox core system architecture (Kobo Toolbox, Harvard Humanitarian Initiative, Boston, Massachusetts, USA). User accounts were password-protected, and data sent to the server was encrypted in transit using SHA256 with RSA encryption that met the data security requirements. Personally identifiable information was not collected, and all records were anonymised with unique study numbers. Study instruments for ANC clients were translated from English into 10 languages (French, Somali, Swahili, Soussou, Poular, Malinké, Keiyo, Maasai, Marakwet and Tugen) by research team members in consultation with language experts, while those for ANC providers and clinic managers were translated into two languages (French and Somali). No back translation was performed. Field data collectors and their supervisors spoke the languages in which the questionnaires were administered. Data collection teams participated in a standardised training with WHO/HRP and the research institutions in each country. The level two intervention was implemented by master trainers in each country who had been trained remotely over a 3-day period following the WHO PCC for FGM prevention facilitator's manual.

### Outcomes

The primary study outcome was the delivery of the 'ABCD' approach by ANC providers measured by responses from their client using tools developed for this study based on previously validated instruments, including four constructs of the operational definition of PCC.[23] We also assessed ANC provider delivery of FGM care services and their utilisation of the level one intervention components. Health facility preparedness to offer FGM prevention and care was assessed using a composite score developed for this study (online supplemental file 2). The secondary self-efficacy outcome was assessed based on a score calculated from a validated tool for measuring general self-efficacy[24] while knowledge, attitudes and practice (KAP) on FGM prevention and care were measured using an unvalidated KAP questionnaire similar to one used in formative research in Guinea. Study instruments can be found in online supplemental file 3.

### Statistical analysis

To have sufficient power (80%) to detect a difference (significance level 5%) between intervention and control arms on the primary study outcome of delivery of the PCC intervention for FGM prevention, 180 ANC clinics,

equally divided across the three study countries were recruited and randomised with 1800 new ANC clients (10 per clinic) recruited at baseline and 1800 at 6-month follow-up. While similar interventions have resulted in a 20% difference between groups,[25] a 10% difference (based on an assumed 20% in the control arm and 30% in the intervention arm) was applied to ensure sufficient power to detect a 10% difference and considering the minimal levels of clinical efficacy for such an intervention to be practical. This sample size also allowed for a 10% non-response and/or loss to follow-up rate and accounted for a clustering effect (intracluster correlation coefficient (ICC)=0.20) at the clinical level. A relatively high level of clustering was assumed in the sample size calculations to not underestimate the needed sample size. Region/county level was not included in the multilevel model due to the low number of included regions/counties per country (Kenya 3, Guinea 2, Somalia 3) and since it would then not be possible to get an accurate estimate of the variance between clusters.

Data were analysed using Stata V.17 software following a per-protocol approach. Data from ANC providers and their clients were analysed if the clinic had at least one provider with follow-up data at all study time points, and in the intervention arm, if the ANC provider present had undergone training on PCC for FGM prevention at month 3. Clinics where providers were lost to follow-up were not included in the final analyses. All facility checklists and ANC client exit interviews were conducted as intended except at sites not accessible due to security issues or closed or converted for care of COVID-19 patients during the pandemic. As the study was designed to pre-screen ANC providers at baseline and include in the final analytical sample only those clinics and providers who were available at 3 and 6 months, an intention-to-treat approach was not feasible. Key characteristics of the participating facilities, providers and clients were summarised. Providers and clinics that were screened but not eligible are compared in online supplemental file 4.

Continuous variables are presented using mean values, and SD while categorical variables are summarised as counts (N) with percentages (%). Differences in proportions were analysed for dichotomous outcomes using Fischer's exact test. For outcomes measured as summary scores, comparisons of mean scores are presented across study arms using t-test.

Initial analyses showed that the clustering was negligible at the provider level since most sites only included one provider in the study. Therefore, multilevel regression models were not used to compare outcomes among providers in intervention versus control arms. However, analyses based on client-level outcomes applied multilevel mixed-effect logistic regression models to assess differences between the study arms. Multilevel analyses were attempted for the models in which ANC clients reported on provider actions, but given the complexity of the models, convergence problems arose leading to unreliable results. In these cases, the results of ordinary models

are presented. Linearity was assessed for the continuous covariates included in the regression models using the Box-Tidwell test in Stata.

At month 6, a comparison of study outcomes between the intervention and control arms was used to determine the combined effect of both levels of the intervention package. Multilevel multivariable logistic regression analyses for ANC provider outcomes were adjusted for their sex, years of service, FGM status, FGM-related training, any specific training on communication/counselling and PCC and whether the provider had conducted FGM in the past. Analyses related to ANC client outcomes were adjusted for their age, educational level, FGM status and exposure to level one IEC materials. These variables were determined a priori based on previously published literature. Analyses related to provider actions as reported by clients were adjusted for client characteristics as it was not possible to definitively link a client with a particular provider. Unadjusted analyses are presented for outcomes that relate to composite measures based on ANC provider and client responses (eg, provision of FGM prevention and care services).

To determine the separate effect of the two levels of the intervention package, additional subanalyses were restricted to the intervention arm. Changes from baseline to month 3 within the intervention arm were used to determine the effect of the level one intervention component while changes from month 3 to month 6 within the same study arm were used to determine the effect of the level two intervention component. The study was not powered for these subanalyses, however, and these results are presented in online supplemental file 4.

In-country data managers monitored data quality. Periodic data audits were conducted by the WHO/HRP quantitative assessment and data management team to identify any data collection gaps and data discrepancies requiring follow-up by in-country teams. Weekly data monitoring meetings were held between the in-country research teams and WHO/HRP staff during data collection periods to identify, document and resolve any data discrepancies. These were virtual due to the COVID-19 pandemic. Given that there was no prospective follow-up of clients, a Data Safety and Monitoring Board was not established. Instead, local research teams documented and reported any unintended harms and/or protocol deviations to the WHO/HRP study coordination team.

### Patient and public involvement statement

Healthcare providers and members of communities where the practice of FGM is prevalent in the study countries were actively involved in the design and implementation of this study intervention. This included the formative research conducted in Guinea, which identified healthcare providers as integral members of FGM practicing communities who understand local community beliefs and norms, making them potential change agents. The formative research also found that the health sector can support these healthcare providers to be effective

change agents by incorporating FGM content within their training, ensuring accountability to legal and policy standards and promoting FGM abandonment as part of a multisectoral approach. Based on this formative work, the PCC training was developed and subsequently piloted among ANC providers in Kenya before being rolled out as part of the multicountry study.

Additionally, the research partners in Guinea, Kenya and Somalia actively engaged with healthcare providers, community members, and community health volunteers to promote ANC among pregnant women in the study counties and to inform them of the study. This strategy was introduced given the reduced ANC uptake during the COVID-19 pandemic. Prior to recruitment, healthcare providers and pregnant women were provided with information about the study, including the burden of the intervention in terms of time, any risks involved in their participation, and the voluntary nature of their participation; they were recruited after providing informed consent.

At present, study dissemination meetings have been conducted in Kenya and Guinea led by the MoH with participation of other stakeholders, including healthcare providers and community members where the study was implemented. The in-country research partners developed policy briefs summarizing country-specific results relevant to national evidence needs, policy development and practice.

### Role of the funders
Apart from WHO/HRP, the study funders had no role in study design or implementation. WHO/HRP, in collaboration with in-country research teams, developed the study protocol, provided data management and analytical support and contributed to interpretation and manuscript writing. WHO/HRP coordinated the successful implementation of this study. The data collection platform was developed and maintained by an outsourced vendor (First Data, Kenya); data management was coordinated by the local implementing partners (CERREGUI, DARS and University of Nairobi) and statistical data analysis was conducted by an external statistician (Dr Max Petzold, Gothenburg University). All these functions were conducted with utmost integrity following the International Council for Harmonisation of Technical Requirements for Pharmaceuticals for Human Use Good Clinical Practice(ICH-GCP) guidelines.

## RESULTS
### Recruitment and retention
Between August 2020 and September 2021, a total of 180 ANC clinics (ie, 60 clinics per study country) were enrolled and randomised to intervention and control arms. There was some natural staggering of the start and subsequent data collection dates due to factors, such as weather, COVID-19, Ramadan and national elections. Data collection periods ranged from 3 to 6 weeks in each country at each time point. The time elapsed between the end of one data collection period to the beginning of the next data collection period ranged from 3 to 5 months.

At month 0, in the intervention arm, 216 providers and 900 clients (ie, 10 per clinic) were interviewed. Based on a review of the clinical rotation schedule to ensure the participation of at least one provider from each study clinic throughout the trial, 133 providers were enrolled in the intervention arm. In the control arm, 220 providers and 900 clients were interviewed (figure 1). At month 3, data were collected at 98% (n=88) of the intervention clinics as two clinics in Kenya were inaccessible due to insecurity. 130 (98%) ANC providers (at least 1 from each site) and 880 first visit ANC clients completed the month 3 questionnaires prior to implementing the level two PCC intervention. No data collection was conducted at the control sites. At month 6, 91% (n=163) of ANC clinics (81, intervention and 82, control) had at least one ANC provider (intervention n=110 and control n=122) on duty

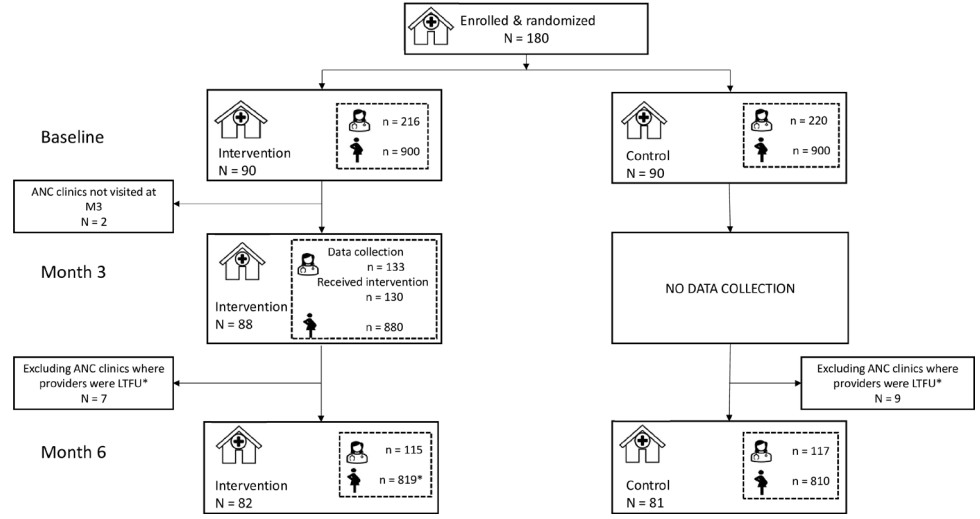

**Figure 1** Study Consolidated Standards of Reporting Trial diagram. ANC, antenatal care; LTFU, loss-to-follow up.

**Table 1** Characteristics of ANC clinics included in month 6 analyses

| Characteristics | Overall (n=163*) | Intervention (n=82) | Control (n=81) |
|---|---|---|---|
| Number of ANC providers | Mean 4 (SD: 3) median 3 (1–14, IQR 3) | Mean 4 (SD: 2) median 3 (1–11, IQR 3) | Mean 4 (SD: 3) median 3 (1–14, IQR 4) |
| Average number of ANC clients/month | Mean 150 (SD: 127) median 118 (3–664, IQR 141) | Mean 148 (SD: 121) median 117 (3–500, IQR 143) | Mean 152 (SD: 133) median 120 (3–664, IQR 140) |
| MoH supervisory visits in the past year | Mean 4 (SD: 3) median 3 (0–18, IQR 2) | Mean 4 (SD: 3) median 4 (1–18, IQR 1) | Mean 4 (SD: 3) median 3 (0–12, IQR 2 |
| Size of catchment population served | Mean 36 754 (SD: 126 082) median 15 972 (1000–1 458 000 IQR 24 332) | Mean 23 649 (SD: 35 873) median 16 022 (1000–290 000 IQR 22 361 | Mean 50 020 (SD: 174 739) median 15 551 (1000–1 458 000 IQR 25 544 |
| Presence of anti-FGM activities in the catchment area | | | |
| Yes | 74 (45%) | 43 (52%) | 31 (38%) |
| No | 89 (55%) | 39 (48%) | 50 (62%) |
| Presence of pro-FGM activities in the catchment area | | | |
| Yes | 21 (13%) | 12 (15%) | 9 (11%) |
| No | 140 (86%) | 68 (83%) | 72 (89%) |
| Do not know | 2 (1%) | 2 (2%) | 0 (0%) |

*Total of 17 ANC clinics not included: 16 clinics were excluded (7 intervention and 9 control) due to loss-to-follow up of ANC provider, that is, the clinics did not have at least 1 ANC provider present across all study time points while 1 ANC clinic in Kenya was never visited at subsequent time points due to issues with insecurity. An ANC provider from one of the clinics in Kenya that had been inaccessible due to insecurity attended the person-centred communication training and was subsequently interviewed.
ANC, antenatal care; FGM, female genital mutilation; MoH, Ministry of Health.

who was previously enrolled in the study. The client questionnaire was completed by 819 and 810 first visit ANC clients, respectively, in the intervention and control sites.

### Characteristics of study sites and participants

The 163 ANC clinics retained to the end of the study, had a mean of 4 ANC providers (SD: 3) and served on average 150 new ANC clients per month (SD: 127) with a mean catchment population of 36 754 people (SD: 126 082). In 55% (n=89) of clinics, the clinic manager reported that there were no activities promoting FGM prevention in the facilities' catchment area (table 1). These characteristics were not different from the 17 ANC clinics that were enrolled at baseline but that subsequently were not included in the final analysis (online supplemental file 4).

Of the 232 ANC providers who contributed data for analysis at month 6, 83% (n=193) were women and their mean age was 36 years (SD: 10 years). They had an average of 8 years of professional experience (SD: 7 years) and 68% (n=158) had studied up to diploma level (generally 3 years post-secondary education) with 90% (n=208) identifying as either midwives, nurses or nurse-midwives. Health cadres were defined by national licensing requirements in each country. Among these providers, at baseline, 63% (n=146) reported that they had not received formal clinical training on FGM prevention and care (table 2). Almost two-thirds (64%, n=149) reported that they had received training on communication/counselling while half (51%, n=118) had received training on

person-centred care. Further, 54% (n=126) of female providers reported that they had undergone FGM while overall, 93% (n=217) of providers reported that they had never performed FGM. These characteristics were not different when compared with the ANC providers who were on duty in the 180 ANC clinics enrolled at baseline but who did not complete the intervention (online supplemental file 4). The mean age of the 1800 clients exiting their first ANC visits at baseline was 26 years (SD: 6 years), 47% (n=846) reported not having received any education and 73% (n=1320) reported that they had undergone FGM. These characteristics were similar to the 880 and 1630 first visit ANC clients interviewed at month 3 (intervention arm only) and month 6, respectively (table 3).

To evaluate potential bias from a differential selection of providers receiving the intervention, we assessed differences in baseline characteristics between the 133 ANC providers from intervention facilities who were screened at baseline and received PCC training at month 3 (ie, included in the analytical sample) versus the 97 who were screened and did not receive the intervention (ie, excluded from analytical sample). The reasons for this exclusion included the fact that some of the providers had been transferred from the study clinics or could not be released to attend the training so as not to affect service delivery. Both groups were similar in terms of sex, educational level, professional cadre, as well as whether they had undergone or recently performed FGM. However,

**Table 2** Characteristics of ANC providers included in the month 6 analyses

| Characteristics | Overall (n=232) | Intervention (n=115) | Control (n=117) |
|---|---|---|---|
| Age | Mean 36 (SD: 10) median 34 (20–65, IQR 15) | Mean 35 (SD: 10) median 33 (20–59, IQR 14) | Mean 37 (SD:11) median 35 (20–65, IQR 16) |
| Years of professional experience | Mean 8 (SD: 7) median 6 (1–39, IQR 7) | Mean 8 (SD:7) median 6 (1–30, IQR 8) | Mean 8 (SD:7) median 6 (1–39, IQR 7) |
| Sex | | | |
| Female | 193 (83%) | 95 (83%) | 98 (84%) |
| Highest educational level | | | |
| Certificate | 21 (5%) | 12 (10%) | 9 (8%) |
| Diploma | 158 (68%) | 72 (63%) | 86 (74%) |
| Bachelors | 44 (19%) | 27 (24%) | 17 (15%) |
| Masters and above | 1 (0.4%) | 0 (0%) | 1 (1%) |
| Other | 8 (3%) | 4 (3%) | 4 (3%) |
| Current professional role/title | | | |
| Midwife | 103 (44%) | 53 (46%) | 50 (43%) |
| Nurse | 51 (22%) | 25 (22%) | 26 (22%) |
| Nurse-midwife | 54 (23%) | 27 (24%) | 27 (23%) |
| Other | 24 (10%) | 10 (9%) | 14 (12%) |
| Received formal training on FGM during clinical training | | | |
| Yes | 85 (37%) | 44 (38%) | 41 (35%) |
| No | 146 (63%) | 71 (62%) | 75 (64%) |
| Do not know | 1 (0.4%) | 0 (0%) | 1 (1%) |
| Timing of clinical training on FGM | | | |
| Pre-service | 33 (14%) | 18 (16%) | 15 (13%) |
| In-service | 45 (19%) | 22 (19%) | 23 (20%) |
| Both pre-service and in-service | 7 (3%) | 4 (4%) | 3 (3%) |
| Received formal training on communication/counselling | | | |
| Yes | 149 (64%) | 76 (66%) | 73 (62%) |
| No | 83 (36%) | 39 (34%) | 44 (38%) |
| Received formal training on person-centred care | | | |
| Yes | 118 (51%) | 58 (50%) | 60 (51%) |
| No | 113 (56%) | 56 (49%) | 57 (49%) |
| Do not know | 1 (0.4%) | 1 (1%) | 0 (0%) |
| Undergone FGM | | | |
| Yes | 126 (54%) | 65 (57%) | 61 (52%) |
| No | 63 (27%) | 27 (24%) | 36 (31%) |
| Do not know | 2 (1%) | 2 (2%) | 0 (0%) |
| Refused to answer | 2 (1%) | 1 (1%) | 1 (1%) |
| Conducted FGM | | | |
| Yes | 15 (7%) | 9 (8%) | 6 (5%) |
| Conducted FGM on a girl <18 years | | | |
| Yes | 14 (6%) | 8 (7%) | 6 (5%) |

ANC, antenatal care; FGM, female genital mutilation.

**Table 3** Characteristics of ANC clients interviewed at each time point

| Characteristics | ANC clients interviewed at baseline | | | ANC clients interviewed at month 3 | ANC clients interviewed at month 6 | | |
| --- | --- | --- | --- | --- | --- | --- | --- |
| | Overall (n=1800) | Intervention (n=900) | Control (n=900) | Intervention only (n=880) | Overall (n=1759) | Intervention (n=879) | Control (n=880) |
| Age | Mean 26 (SD: 6) median 25 (15–45, IQR 10) | Mean 25 (SD: 6) median 25 (15–45, IQR 10) | Mean 26 (SD: 6) median 25 (15–45, IQR 10) | Mean 26 (SD 6) median 25 (15–45, IQR 10) | Mean 26 (SD: 6) median 25 (15–45, IQR 9) | Mean 26 (SD: 6) median 25 (15–45, IQR 9) | Mean 26 (SD: 6) median 25 (15–45, IQR 10) |
| Highest educational level | | | | | | | |
| None | 840 (47%) | 407 (45%) | 433 (48%) | 439 (50%) | 806 (46%) | 384 (44%) | 422 (47%) |
| Primary | 484 (27%) | 231 (26%) | 253 (28%) | 239 (27%) | 553 (31%) | 278 (32%) | 275 (31%) |
| Secondary | 331 (18%) | 171 (19%) | 160 (18%) | 157 (18%) | 306 (17%) | 160 (18%) | 146 (16%) |
| University | 95 (5%) | 61 (7%) | 34 (4%) | 25 (3%) | 67 (4%) | 34 (4%) | 33 (4%) |
| Other | 50 (3%) | 30 (3%) | 20 (25) | 20 (2%) | 37 (2%) | 23 (3%) | 14 (2%) |
| Have you undergone FGM? | | | | | | | |
| Yes | 1320 (73%) | 677 (75%) | 643 (71%) | 645 (73%) | 1321 (75%) | 655 (75%) | 666 (75%) |
| No | 452 (25%) | 209 (23%) | 243 (27%) | 224 (25%) | 420 (24%) | 206 (23%) | 214 (24%) |
| Do not know | 12 (1%) | 10 (1%) | 2 (0.2%) | 5 (1%) | 21 (1%) | 13 (2%) | 8 (1%) |
| Refused to answer | 16 (1%) | 4 (0.4%) | 12 (1%) | 6 (1%) | 7 (0.4%) | 5 (1%) | 2 (0.2%) |

ANC, antenatal care; FGM, female genital mutilation.

included providers tended to be slightly younger (by 2 years on average) and less likely to be of the Muslim religion, although the question on religion was not administered for the Somalia sample since all respondents were assumed to be Muslim (online supplemental file 4).

## ANC providers implementation of ABCD elements of the PCC approach

Table 4 presents the analysis of study outcomes by arm at month 6. Compared with ANC providers in the control arm, those in the intervention arm were nearly nine times as likely to ask their clients if they had undergone FGM (adjusted OR (AOR): 8.9, 95% CI: 6.9 to 11.5; p<0.001), nearly 10 times as likely to ask their clients' personal beliefs regarding FGM (AOR: 9.7, 95% CI: 7.5 to 12.5; p<0.001), more than nine times as likely to discuss with their clients why FGM should be prevented (AOR: 9.2, 95% CI: 7.1 to 11.9; p<0.001) and nearly eight times as likely to discuss with their clients how FGM could be prevented (AOR: 7.7, 95% CI: 6.0 to 9.9; p<0.001). Further, ANC clients in the intervention arm were nearly seven times as likely to report that they were satisfied with how FGM had been addressed by their provider during the clinical visit compared with those in the control arm (AOR: 6.6, 95% CI: 5.1 to 8.4; p<0.001). In the intervention arm, the mean score of implementing the ABCD elements of the PCC approach was more than two times higher (p<0.001) in the intervention (3.9 (3.8–4.0)) compared with the control arm (1.6 (1.5–1.7)).

## ANC clinical preparedness to provide FGM prevention and care services

A significantly higher proportion of ANC clinics in the intervention arm had all correct responses related to facility preparedness to provide FGM prevention and care services compared with those in the control arm (68% vs 27%, p<0.001). Additionally, ANC clinics in the intervention arm had a significantly higher mean score for preparedness compared with those in the control arm (3.4 (95% CI: 3.2 to 3.6) vs 2.6 (95% CI: 2.4 to 2.9; p<0.001)).

## ANC providers using level one intervention components

A higher proportion of ANC providers in the intervention arm reported having used the level one intervention package components compared with those in the control arm (92% vs 56%, p<0.001). In multivariable analyses, ANC providers in the intervention arm were ten times as likely to report having used the level one intervention package components compared with those in the control arm (AOR: 10.1, 95% CI: 4.6 to 22.4; p<0.001).

## ANC providers offering appropriate FGM prevention and care services

At month 6, based on a cumulative score to specific questions on the provision of appropriate FGM-related prevention and care services, ANC providers in the intervention arm had higher scores than those in the control

**Table 4** Analysis of study outcomes

**Primary Outcomes**

**ANC clients reporting that their provider implemented components of PCC for FGM prevention approach**

|  | Intervention (n=819) | Control (n=810) | Adjusted OR* § (95% CI) | P value | ICC |
|---|---|---|---|---|---|
| Provider asked client if they have undergone FGM | 634 (77%) | 245 (30%) | 8.9 (6.9 to 11.5) | <0.001 | N/A |
| Provider asked client about the client's personal beliefs regarding FGM | 616 (75%) | 217 (27%) | 9.7 (7.5 to 12.5) | <0.001 | N/A |
| Provider discussed with client why FGM should be prevented | 629 (77%) | 244 (30%) | 9.2 (7.1 to 11.9) | <0.001 | N/A |
| Provider discussed with client how FGM could be prevented | 592 (72%) | 232 (29%) | 7.7 (6.0 to 9.9) | <0.001 | N/A |
| Client satisfied with how FGM was addressed by provider during clinical visit | 684 (84%) | 348 (43%) | 6.6 (5.1 to 8.4) | <0.001 | N/A |
|  |  |  | **Difference in mean scores (95% CI)** |  |  |
| Mean score of implementing PCC approach (out of 5) | 3.9 (3.8–4.0) | 1.6 (1.5–1.7) | 2.3 (2.1 to 2.5) | <0.001 | N/A |
| Mean score of PCC+appropriate FGM prevention and care (out of 8) | 6.2 (5.9–6.6) | 3.7 (3.2–4.1) | 2.6 (2.0 to 3.2) | <0.001 | N/A |

**ANC clinical preparedness to offer FGM prevention and care services**

|  | Intervention (n=82) | Control (n=81) |  | P value | ICC |
|---|---|---|---|---|---|
| Clinics with ALL correct responses for preparedness | 56 (68%) | 22 (27%) | – | <0.001 | N/A |
|  |  |  | **Difference in mean scores (95% CI)** |  |  |
| Mean score of clinical preparedness (out of 4) | 3.4 (3.2–3.6) | 2.6 (2.4–2.9) | 0.8 (0.4 to 1.1) | <0.001 | N/A |
|  | **Intervention (n=115)** | **Control (n=117)** | **Adjusted OR* ‡ (95% CI)** | **P value** | **ICC** |
| Providers using level one intervention package | 106 (92%) | 65 (56%) | 10.1 (4.6 to 22.4) | <0.001 | N/A |

**Secondary outcomes**

|  | Intervention (n=115) | Control (n=117) | Adjusted OR*‡ (95% CI) | P value | ICC |
|---|---|---|---|---|---|
| Providers with appropriate interpersonal communication skills | 82 (71%) | 68 (58%) | 1.8 (1.0 to 3.2) | 0.035 | N/A |
| Providers with high self-efficacy | 93 (81%) | 99 (85%) | 0.7 (0.3 to 1.4) | 0.317 | N/A |
| Providers reporting less supportive attitudes towards FGM | 84 (73%) | 85 (73%) | 1.0 (0.5 to 1.8) | 0.993 | N/A |
| Providers with high confidence scores¶ | 113 (98%) | 104 (89%) | 7.0 (1.5 to 32.3) | 0.012 | N/A |
|  |  |  | **Unadjusted OR (95% CI)** |  |  |
| Providers not supportive of FGM | 110 (96%) | 114 (97%) | 0.7 (0.2 to 3.3) | 0.677 | N/A |
| Providers not supportive of medicalised FGM | 114 (99%) | 116 (99%) | 1.0 (0.1 to 15.9) | 0.990 | N/A |
| Providers with correct FGM-related knowledge responses | 9 (8%) | 1 (1%) | 9.8 (1.2 to 79.0) | 0.031 | N/A |
|  |  |  | **Difference in mean scores (95% CI)** |  |  |
| Mean score of FGM-related knowledge (out of 6) | 2.5 (2.2–2.7) | 1.6 (1.5–1.8) | 0.8 (0.5–1.1) | <0.001 | N/A |

**Table 4** Continued

**Other ANC client outcomes**

| | Intervention (n=819) | Control (n=810) | Adjusted OR†§ (95% CI) | P value | ICC |
|---|---|---|---|---|---|
| Clients reporting less support for FGM after ANC clinical visit | 424 (52%) | 237 (29%) | 5.4 (2.4 to 12.4) | <0.001 | 0.66 |
| Clients reporting that they were strongly opposed to FGM | 498 (61%) | 382 (47%) | 2.4 (1.1 to 5.2) | 0.023 | 0.62 |
| Clients reporting that they intend to have their daughters cut | 96 (12%) | 209 (26%) | 0.3 (0.1 to 0.7) | 0.004 | 0.6 |
| Clients reporting that they would prefer healthcare provider to cut daughters | 53 (7%) | 139 (17%) | 0.2 (0.1 to 0.5) | <0.001 | 0.54 |
| Clients wishing to be active in FGM prevention | 677 (83%) | 535 (66%) | 3.2 (1.6 to 6.2) | 0.001 | 0.5 |

*Single-level multivariable adjusted models.
†Multilevel multivariable adjusted models.
‡Provider outcomes adjusted for sex, years of service, FGM status, FGM-related training, any specific training on communication/counselling and PCC and whether the provider had conducted FGM in the past.
§Client outcomes adjusted for age, educational level, FGM status and exposure to level one information, education and communication materials.
¶This analysis includes 217 observations instead of 232 because of missing data on some covariates
ANC, antenatal care; FGM, female genital mutilation; ICC, intracluster correlation coefficient; OR, Odds ratio; PCC, person-centred communication .

arm with a difference in mean score of 2.6, 95% CI: 2.0 to 3.2; p<.001.

### ANC providers' confidence, self-efficacy and communication skills

Providers in the intervention arm had significantly better interpersonal communication skills compared to providers in the control arm (AOR: 1.8, 95% CI: 1.0 to 3.2; p=.035). A higher proportion of ANC providers in the intervention arm reported being confident in their knowledge to provide FGM prevention and care services compared with those in the control arm (98% vs 89%, p=0.005). In multivariable analysis, ANC providers in the intervention arm had seven times the odds of reporting being confident in their knowledge to provide FGM prevention and care services compared with those in the control arm (AOR: 7.0, 95% CI: 1.5 to 32.3; p=0.012). Self-efficacy was generally high with no significant difference between study arms in having high scores (81% vs 85%, p=0.36 and AOR: 0.7, 95% CI: 0.3 to 1.4; p=0.317).

### ANC providers' knowledge, attitudes and support for FGM/ medicalised FGM

The mean correct scores out of 6 for FGM-related knowledge were higher among ANC providers in the intervention arm compared with the control arm (2.5, 95% CI: 2.2 to 2.7 vs 1.6, 95% CI: 1.5 to 1.8; p<0.001) but 8% versus 1% (p=0.01) had correct responses on all of the FGM-related knowledge questions, showing low knowledge overall and particularly on FGM typology. ANC providers in the intervention arm had nearly ten times the odds of having correct FGM-related knowledge than those in the control arm (OR: 9.8, 95% CI: 1.2 to 79.0; p=0.031). Providers in both groups had similarly unsupportive attitudes towards

FGM and similarly unsupportive attitudes about medicalised FGM with most providers reporting that they did not support FGM (96% vs 97%, p=0.677) and/or medicalised FGM (99% vs 99%, p=0.90).

### ANC clients' support for FGM, intention to have their daughters undergo FGM and being involved in FGM prevention efforts

Compared with those in the control arm, a higher proportion of ANC clients in the intervention arm reported being less supportive of FGM after their month 6 clinical visit (52% vs 29%, p<0.001). In multivariable analysis, ANC clients in the intervention arm had more than twice the odds of reporting that they were strongly opposed to FGM (AOR: 2.4, 95% CI: 1.1 to 5.2; p=0.023, ICC: 0.62). When asked about their support for FGM, clients in the intervention arm compared to the control arm had more than five times the odds of being less supportive of FGM after their clinic visit (AOR: 5.4, 95% CI: 2.4 to 12.4; p<0.001, ICC: 0.66). ANC clients in the intervention clinics had lower odds of intending to have their daughters undergo FGM (OR: 0.3, 95% CI: 0.1 to 0.7; p=0.004, ICC: 0.60) or of wanting a healthcare provider to perform FGM (AOR: 0.2, 95% CI: 0.1 to 0.5; p<0.001, ICC: 0.54) and higher odds of reporting that they wished to be active in FGM prevention (AOR: 3.2, 95% CI: 1.6 to 6.2, p=0.001, ICC: 0.50).

To understand the impact of the level one intervention relative to the level two intervention, a comparison of study outcomes restricted to the intervention arm was done between baseline and month 3 and between months 3 and 6 (online supplemental file 4). Although not statistically powered for this analyses, we found that

a significantly higher proportion of ANC clients in the intervention arm reported that their provider had asked about the different PCC components at month 3 versus baseline and at month 6 versus month 3. Similarly, a significantly higher proportion of ANC clinics were prepared to provide FGM-related prevention and care services at month 3 compared with baseline and at month 6 compared with month 3. No statistically significant differences were seen in the proportion of ANC providers with the secondary outcomes apart from high confidence scores observed between month 6 and month 3. Finally, ANC client outcomes were significantly higher among intervention clients in month 3 versus baseline and in month 6 versus month 3.

## DISCUSSION

The results of this cluster randomised trial show that an intervention to strengthen health facility preparedness while building skills of ANC providers to communicate using a person-centred counselling technique on FGM prevention was effective. ANC providers exposed to the intervention had greater confidence, higher FGM-related knowledge scores, and more effective delivery of FGM prevention and care services as compared to those in the control group. Additionally, ANC clients who had received care from these providers were less supportive of FGM and had reduced intentions to perform FGM on their daughters. This study provides evidence of a practical intervention to engage healthcare providers in FGM abandonment efforts while also offering quality care to FGM survivors. This study provides evidence of how to effectively build the capacity of healthcare providers at primary care to address FGM,[26] an area identified as a critical gap during the formative research.

The PCC training modules strengthened ANC providers' skills on FGM prevention and care and helped to clarify their beliefs and attitudes, which are key drivers of FGM.[27] We did not find notable differences in attitudes among ANC providers in the two groups. The knowledge scores, while higher in the intervention group, were low overall, and on further investigation, it appears that questions on typology captured through visually drawn images on a tablet device were consistently answered incorrectly. These results perhaps show measurement and knowledge limitations but do not necessarily relate to service provision or quality of care. Attitudes in the intervention and control groups were generally unsupportive of FGM and did not appear to be heavily impacted by the training intervention. Exposure to the intervention package also did not improve ANC providers' self-efficacy towards FGM prevention and care. This may be related to the lack of support for FGM and/or its medicalisation and high self-efficacy among nearly all providers throughout the study in both study arms, a finding that was also noted in formative research conducted in Guinea.[28 29] In the formative phase, while the vast majority of health workers were opposed to the practice, 38% also felt that FGM

limited promiscuity and 7% believed that it was a good practice, showing ambivalence and complexity in attitudes about FGM among health providers. Other studies have found that some providers support the perpetuation of the practice and even plan to have their own daughters undergo FGM or to perform it on their clients.[30]

The findings in this study underscore the importance of addressing the values and attitudes of both providers and clients as a means of achieving positive behavioural change. Changes observed among ANC providers were sustained across the study duration and ultimately, and importantly, resulted in reported changes in the attitudes and intentions of their clients. However, this study design did not allow us to determine whether the attitudinal changes observed among ANC clients were sustained after their clinical visit or translated into positive change in FGM prevention.

The application of these study results into programming will need to consider several factors. First, the study sites were primary care facilities located in high FGM prevalence settings. The results of this intervention may not be generalisable to settings where FGM is less prevalent or to settings other than primary care. Second, first ANC visits are not typical of other health visits since the consultation is generally longer with a greater focus on health promotion messaging. While this is an ideal setting for implementing such an intervention, its application to other health settings and among other population groups is not known. During scale-up, if the PCC approach is applied among clients seeking other sexual and reproductive health services or parents bringing their children to child immunisation and wellness visits, it will be important to consider time requirements for the delivery of the 'ABCD' steps, especially in high volume clinical settings.

Third, while the study found a positive impact of the PCC training on healthcare providers' delivery of person-centred FGM prevention counselling, the continuity and quality of FGM prevention counselling in the long-term is not known. Specifically, it will be important to assess subsequently whether providers will continue to provide prevention counselling on an ongoing basis, whether they will share their learnings with family and community members and whether clients will follow through with their intentions to not have their daughters undergo FGM. It may be important to include a supervisory mentorship component to ensure implementation of this intervention[31] in order to strengthen PCC communication practice and quality.

### Limitations

The implementation of this multicountry study was not without challenges and limitations. First, initiation of field data collection activities was delayed by the global COVID-19 pandemic in 2020–2021 and required some modification to trainings of the data collection teams, the master trainers and the ANC providers receiving the PCC

intervention. This may have impacted the overall effectiveness of the intervention.

Second, to attempt to ensure the participation of at least one provider at each site, all providers were pre-screened at baseline and clinical rotation schedules determined enrolment into the study. Selection bias might have been introduced through this process. The exploratory analysis to assess for selection and attrition bias from the pre-screen step, did not reveal significant differences between included and excluded health workers except for slightly lower age (online supplemental file 4), and a per-protocol analysis was required, but it is possible that differences in other unmeasured factors related to the clinics and providers might have biased the results. Findings from a process evaluation conducted as part of this study provide additional insights on the feasibility, acceptability, appropriateness and fidelity of the intervention implementation in these contextual settings to inform further implementation and scale-up.[32]

Third, we did not perform adjustment for multiple testing in our analysis given that the different tests are interpreted separately and no overall conclusion will be stated. Given that the null hypotheses of no differences are true, we estimate that the overall type one error rate is higher than the individual test level of 0.05. In terms of assumptions regarding clustering, sample size was calculated based on an ICC of 0.20. However, the observed ICCs were all above 0.50 leading to statistically conservative conclusions of the non-significant results due to being underpowered to find an association.

Finally, we acknowledge that there are many factors that could impact FGM-related decision-making and a positive and impactful interaction with a respected healthcare provider might not be sufficient to lead to actual changes in community behaviour. However, the study design enabled us to compare similar sites to identify the relative effect of this approach since both intervention and control sites would be exposed to similar factors, and clients at these sites would face similar complexities in decision-making.

## CONCLUSION

In conclusion, this study highlights the importance of addressing the values and beliefs of healthcare providers working at the primary care level, who are subject to social norms around FGM that may conflict with medical ethics and national laws and policies, as an intermediary step in preventing FGM. Empowering these healthcare providers with communication skills and engaging them as opinion leaders can be impactful in changing their clients' attitude towards FGM. In conjunction with FGM prevention activities in other sectors, this intervention can contribute to positive change if brought to scale.

**Author affiliations**
[1]Centre for Research in Reproductive Health in Guinea, Conakry, Guinea
[2]Department of Obstetrics and Gynecology, University of Nairobi, Nairobi, Kenya
[3]Department of Sexual and Reproductive Health and Research, World Health Organization, Geneva, Switzerland
[4]Amref International University, Nairobi, Kenya
[5]Department of Biochemistry and Biotechnology, Technical University of Kenya, Nairobi, Kenya
[6]United Nations Population Fund, Addis Ababa, Ethiopia
[7]Division of Healthier Populations, World Health Organization, Geneva, Switzerland
[8]Public Health and Community Medicine, University of Gothenburg Sahlgrenska Academy, Gothenburg, Sweden
[9]Central Statistics Department, Ministry of Planning and National Development, Hargeisa, Somaliland, Somalia
[10]Data and Research Solutions, Hargeisa, Somaliland, Somalia

**Acknowledgements** The authors would like to acknowledge the funders, the WHO country office colleagues, Dr Bernadette Dramou, Dr Cécé-Vieux Kolie, Dr Joyce Lavussa, Ms Matilda Cherono, Ms Asia Ahmed Osman in the study countries as well as MoH and national FGM stakeholders in the development of the research protocol, implementation of study interventions and field data collection. We also thank Professor Joanna Schellenberg of the London School of Hygiene and Tropical Medicine and Dr Christina Atchison of Imperial College London for their input in conceptualising the study as well as Dr Leyla Hussein for supporting pilot testing of the PCC intervention.

**Contributors** WA and CP conceptualised the study and prepared the protocol in collaboration with VM, KS, PMN, TE, MDB, A-MS, ADia and MAA. MDB, A-MS, AOS, PMN, TE, JMK, ADia and MAA provided oversight over study implementation while ADia and JK monitored data quality in countries and KN and SST monitored data quality across countries. VM prepared the first draft of the manuscript with input from WA and CP, the responsible officer of the study at WHO/HRP. MP developed the statistical analysis plan and conducted data analysis. KS coordinated the development of the PCC for FGM prevention training. KS, PMN, TE, JMK, JK, MDB, A-MS, AOS, ADia, ADir, SA and MAA contributed to and reviewed the manuscript for proper intellectual content. CP is the guarantor and accepts responsibility for the overall content of the manuscript. All authors read and approved the final draft of this manuscript.

**Funding** This work received funding from the Governments of Norway and the UK of Great Britain and Northern Ireland as well as the UNDP-UNFPA-UNICEF-WHO-World Bank Special Programme of Research, Development and Research Training in Human Reproduction (HRP), a cosponsored programme executed by the WHO.

**Disclaimer** The author is a staff member of the World Health Organization. The author alone is responsible for the views expressed in this publication and they do not necessarily represent the views, decisions or policies of the World Health Organization. The named authors alone are responsible for the views expressed in this publication and do not necessarily represent the decisions or the policies of the UNDP-UNFPA-UNICEF-WHO-World Bank Special Programme of Research, Development and Research Training in Human Reproduction (HRP) or the WHO.

**Competing interests** None declared.

**Patient and public involvement** Patients and/or the public were involved in the design, or conduct, or reporting, or dissemination plans of this research. Refer to the Methods section for further details.

**Patient consent for publication** Consent obtained directly from patient(s).

**Ethics approval** This study involves human participants and was approved by (1) WHO Ethical Review Committee (ERC) (#P151/03/2014). (2) Kenya: Kenyatta National Hospital/University of Nairobi ERC (P805/09/2019) and the National Commission for Science, Technology, and Innovation (NACOSTI/P/20/5721) (3) Somalia: the Department of Planning, Policy and Strategic Information, Unit of Research (MOHD/DG: 2/11526/2019) (4) Guinea: the Comité National d'Ethique Pour la Recherche en Santé (CNERS) (105/CNERS/19). Participants gave informed consent to participate in the study before taking part.

**Provenance and peer review** Not commissioned; externally peer reviewed.

**Data availability statement** Data are available upon reasonable request. De-identified data set will be retained in the WHO HRP electronic archival system. Any use of the de-identified analytic data set for secondary research purposes will be governed by the WHO data use regulation. Request for data dictionary and for data set may be sent to pallittoc@who.int.

**ORCID iD**
Christina Pallitto http://orcid.org/0009-0007-1700-6774

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
