## [Reviewer comments · BMJ Open]

ARTICLE DETAILS

TITLE (PROVISIONAL)	A cluster randomized trial of a health system strengthening approach applying person-centered communication for the prevention of female genital mutilation in Guinea, Kenya, and Somalia
AUTHORS	Balde, Mamadou; Ndavi, Patrick; Oyaró, Vernon; Soumah, Anne-Marie; Esho, Tammary; King'oo, James; Kemboi, Jackline; Sall, Alpha; Diallo, Aissatou; Ahmed, Wisal; Stein, Karin; Nosirov, Khurshed; Thwin, Soe Soe; Petzold, Max; Ahmed, Muna; Diriye, Ahmed; Pallitto, C

VERSION 1 – REVIEW

REVIEWER	Berg, Rigmor Norwegian Knowledge Centre for the Health Services , Dept. of Evidence-Based Summaries
REVIEW RETURNED	18-Nov-2023

GENERAL COMMENTS	There are a number of strengths of this C-RCT, including the study registration, reporting in accordance with CONSORT, surveys available in multiple languages, and a published protocol. In my comments, I focus on areas for improvement. I disagree with the authors that this is a hybrid design, because the study has no clinical outcome data. There are no clinical FGM behavioral data collected. The data collected from providers are knowledge, attitudes, confidence, support, communication skills, whether the provider talked to clients about FGM, whether the provider gave FGM-related care (discourage client to perform FGM, look for presence of FGM in client, record clients' FGM in file). The data collected from clients similarly pertain to knowledge, attitudes, intentions, support. It is unclear what the purpose of the data collection at 3 months is. In the manuscript, only sociodemographic data of clients at month 3 are provided; there's no outcome data. It can be stated, as the authors do, that the trial is phased, but why there was several months between providing the materials (to both intervention and control clinics) and training the providers in the intervention clinics in communication skills, is unclear. Regarding methods, is unclear how the researchers managed to ensure the busiest clinics were randomized to different arms. The study assessed adhering to intervention (the 'per-protocol' effect), not effect of assignment to intervention, and they should make this clearer. It is difficult to understand the reason for why an intent-to-treat evaluation was not possible. Given it is a per-protocol effect assessment, they also need to address any occurrence of non-
--

	protocol interventions, failures in implementing the intervention that could have affected the outcome, and non-adherence to the assigned intervention by trial participants. It should be stated whether/how the allocation sequence was concealed until participants were enrolled and assigned to interventions. I find it problematic that the researchers state that clients and providers were blinded to study arm allocation. Providers cannot be unaware that they are given communication training. It is highly likely that both providers and the clients are aware of their assigned intervention during the trial. It should be stated whether the field data collectors (outcome assessors) were blinded to assigned intervention. Some more information about the intervention itself (training), that differed between the two arms, should also be provided, such as how much/number of hours of training in how to deliver a standardized “ABCD” approach for PCC did the providers receive. It would strengthen the study if the researchers explained their choice of ICC=0.20. With respect to the outcomes, the researchers state that the instruments were validated “where possible”. In principle, it is always possible to perform validation of an instrument. In this manuscript, it should be stated which instruments, measuring which outcomes, were validated and which were not. All outcomes are self-report, either by provider or by client. It is stated that the primary outcome is delivery of the ABCD approach by ANC providers: on page 62-63 we see that it is actually four outcomes (facility preparedness to provide FGM services, provider utilization of WHO handbook of FGM, providing FGM related care (i.e. discourage FGM, look for FGM when examine, record FGM in file), deliver ABCD package (as perceived by clients). I suggest a revision or explanation of this. There are a number of secondary outcomes. The results and tables should be ordered with the main outcome presented first, followed by the secondary outcomes. And the discussion should highlight the result for the main outcome. It is noteworthy that the intervention did not succeed in improving providers’ FGM-related knowledge, self-efficacy, FGM-related attitudes (table3), and these results deserve more discussion. Concerning reporting, it would be useful if there was clearer information about any use of restrictions in connection with the randomization procedure, mechanisms used to implement the random allocation sequence and steps taken to conceal the allocation, who were involved in the allocation sequence (generated allocation sequence, enrolled clinics, assigned clinics). Finally, there appears to have been 3 funders (government of Norway, government of UK, WHO/HRP).One funder, WHO/HRP, appears to have been heavily involved in the study, including design, implementation, protocol, data management and analysis, interpretation and manuscript. The potential for bias is considerable and must be addressed.
--	---

REVIEWER	McConnachie, Alex University of Glasgow, Robertson Centre for Biostatistics
REVIEW RETURNED	18-Feb-2024

GENERAL COMMENTS	Balde et al present the results of a cluster randomised trial of training in person-centred communication in relation to the prevention of female genital mutilation in three countries in Africa. This review considers the use of statistics in the paper.
--

	Generally, these are quite good, though I think a few modifications could be made that would improve the paper. The authors report compliance with the CONSORT guidelines, and provide a checklist in support of this. However, they have used the regular version, not the checklist of cluster randomised trials, which would be more appropriate in this case. There is a section about blinding, though for this sort of trial, I feel that it is important, if possible, for the outcome assessors to be blind to the allocation of the clinic. Given that only intervention clinics were assessed at three months, it may be that this was not possible. Either way, I think it is important to be clear on this point. The study was designed with a pre-specified sample size, which accounted for a healthy dose of clinic-level clustering, which is good. However, I believe a little more detail is required; there is currently not enough information to replicate the calculation (the protocol is the same). In particular, while the authors state that the study was design to have power to detect a 10% difference, the sample size required will depend in the outcome rates in the two groups, not just on the difference. Could the assumed percentages for both groups be made clear? The analyses presented do not take account of clustering, on the basis that any clustering was negligible, and the results from multilevel models were “almost equal”. That sounds reasonable, but I suggest that for the primary analysis paper, it would be better to report the results from multilevel models, or otherwise take account of clustering. If the results are almost equal, it will make no difference to the conclusions. And, it will allow reporting of the ICCs observed in this study, which is one of the recommendations of the CONSORT extension for cluster RCTs. At a minimum, these results ought to be provided as part of the supplementary materials. There is a statement about checking continuous covariates for linearity, but it is not clear how this was done. The term “multiple variable [logistic] regression” is used, though I feel “multivariable” is more commonly used. Under “Characteristics of study sites and participants”, there is a statement that “...characteristics were not different from the 17 ANC clinics that were enrolled at baseline but that subsequently were lost to follow-up.” No evidence is provided in support of this – could a table be added to the supplements to show this, perhaps? There is a similar issue when talking about the ANC providers who did or did not receive PCC training – the differences and similarities between these providers is discussed, but no data are given in support of these statements. In the same section, there is a typo: “(64%, n=14)”; should this be n=149? In the section “Health facility preparedness”, and for much of the subsequent sections, several results are presented, with p-values, though it should be made clear in which table(s) these data are presented.
--	--

	There is a comment that scores for FGM-related knowledge ranged from 1.6 to 2.5, but it is also stated that the mean score in the intervention arm was 2.5, but this would not be consistent with the upper end of the range being 2.5. Perhaps I am misinterpreting something here, so maybe it needs to be clearer. In Table 3, I think it would be easier to read if the intervention and control arm data were given in separate columns. It is not clear why some of the odds ratios are not reported, and, for the continuous outcomes, why there are no estimates of the mean differences, with 95% CIs and p-values. These should be given for all outcomes. Note, given my earlier comments, I feel these should be the intervention effect estimates derived from models allowing for clustering, and the ICCs should be reported – this is important information for others wishing to design trials in the same setting. I feel that Table 3, and the paper as a whole, could make it much clearer what the primary outcome of the study was. In the methods, it is stated that the primary outcome was the delivery of the “ABCD” approach by ANC providers. Since this was as defined in the protocol, it cannot be changed, but it seems an odd choice, since only those in the intervention arm were trained in using this approach. On that basis, it hardly seems like a fair outcome on which to compare the two groups.
--	--

VERSION 1 – AUTHOR RESPONSE

Reviewer: 1

Dr. Rigmor Berg, Norwegian Knowledge Centre for the Health Services

Comments to the Author:

There are a number of strengths of this C-RCT, including the study registration, reporting in accordance with CONSORT, surveys available in multiple languages, and a published protocol. In my comments, I focus on areas for improvement.

I disagree with the authors that this is a hybrid design, because the study has no clinical outcome data. There are no clinical FGM behavioral data collected. The data collected from providers are knowledge, attitudes, confidence, support, communication skills, whether the provider talked to clients about FGM, whether the provider gave FGM-related care (discourage client to perform FGM, look for presence of FGM in client, record clients' FGM in file). The data collected from clients similarly pertain to knowledge, attitudes, intentions, support.

The term hybrid effectiveness implementation research refers to a specific type of implementation research design in which the study seeks to establish efficacy in a real world setting of clinical practice. The hybrid aspect refers to the fact that the study assesses effectiveness of the intervention and the implementation strategy (<https://www.bmj.com/content/347/bmj.f6753.long>) as opposed to other forms of implementation research that assess implementation and/or scale-up strategies of interventions already established as being effective. This study fits the definition of a hybrid-effectiveness implementation research design.

Regarding the point about clinical outcomes, we consider service provision to be an appropriate outcome that aligns with the type of research questions explored in implementation research. Clinical service provision and its effect on intentions of clients are measurable intermediary outcomes towards

more long-term goals of FGM prevention, which do not lend themselves to change in a relatively short time period.

It is unclear what the purpose of the data collection at 3 months is. In the manuscript, only sociodemographic data of clients at month 3 are provided; there's no outcome data. It can be stated, as the authors do, that the trial is phased, but why there was several months between providing the materials (to both intervention and control clinics) and training the providers in the intervention clinics in communication skills, is unclear.

In addition to the month 6 analysis assessing the combined effect of the two levels of the intervention package between both study arms, changes observed from baseline to month 3 within the intervention arm enabled us to explore the effect of the level one intervention component alone. In addition, changes from month 3 to month 6 within the intervention arm enable us to explore the additional effect of the level two intervention component. These analyses showed statistically significant improvements in study outcomes for each intervention component. However, the study was not powered for these analyses and based on feedback from a previous review, we decided to focus this manuscript on the main study analyses. However, since these analyses are included in the study protocol, as the reviewer suggested, we now present them in Appendix 3. The manuscript has also been revised under the Methods section to include this information.

Regarding methods, is unclear how the researchers managed to ensure the busiest clinics were randomized to different arms.

In each study country, facility workload was used as an initial screening criterion for selection of ANC clinics seeing on average 30 new ANC clients per month. The list of included sites was based on data provided by the Ministry of Health in each country. Study facilities were listed in order of client load and then matched into pairs based on this criterion so the two busiest would be randomized to different study arms and the next two and so on. The randomization of pairs was conducted using the uniform random number function in STATA 17.

The study assessed adhering to intervention (the 'per-protocol' effect), not effect of assignment to intervention, and they should make this clearer. It is difficult to understand the reason for why an intent-to-treat evaluation was not possible.

For the facility checklist and ANC client exit interviews, all clinics except for those not accessible due to security issues or closed or converted for COVID-19 care contributed to the analysis. For the ANC provider questionnaire, this study was designed to screen providers and include in the analytic sample only those who would be available at the clinic at 3 and 6 months as a condition of eligibility. For these reasons, an ITT approach was not feasible. This is explained on page 11.

Given it is a per-protocol effect assessment, they also need to address any occurrence of non-protocol interventions, failures in implementing the intervention that could have affected the outcome, and non-adherence to the assigned intervention by trial participants.

The study included two levels of intervention. The Level One intervention consisted of distribution of relevant resources and materials to clinics in both study arms. The implementation of the Level One intervention was independently verified by clinic managers who signed a form confirming receipt of the package at their health centre. In terms of the Level Two intervention, a pre-screening of 436 ANC providers at the intervention and control sites to recruit the providers who would be present at sites considering clinic rotation patterns. As shown in Figure 2, there were 133 providers who were eligible at the intervention sites, and 130 of them were trained at month three.

To evaluate potential bias from the selection of providers we assessed differences in baseline characteristics between ANC providers at intervention facilities who were screened at baseline and received PCC training at month 3 (i.e., included in the analysis sample) versus those who were screened and did not receive the intervention (i.e., excluded from analysis sample). These groups were similar in terms of sex, educational level, professional cadre, as well as whether they had undergone or recently performed FGM, however, included providers tended to be slightly younger (by 2 years on average) and less likely to be of Muslim religion. As the question on religion was not administered for the Somalia sample (all were assumed to be Muslim), selection bias from these differences cannot be assumed. The three providers who did not receive the training but who were eligible were considered missing at random and no selection bias is assumed.

It should be stated whether/how the allocation sequence was concealed until participants were enrolled and assigned to interventions. I find it problematic that the researchers state that clients and providers were blinded to study arm allocation. Providers cannot be unaware that they are given communication training. It is highly likely that both providers and the clients are aware of their assigned intervention during the trial. It should be stated whether the field data collectors (outcome assessors) were blinded to assigned intervention.

Attempts were made to blind clinic managers, ANC providers and their clients to study arm allocation. Since both study arms received the level one intervention component at baseline, and the providers and managers at control sites were unaware of the training that took place at intervention sites, it is conceivable that they were not aware of their study arm. Presumably, intervention clients would assume they were the intervention arm, but they were also not aware of what might have been offered to other sites. ANC clients, however, were completely blinded as to study arm allocation since a distinct set of clients was interviewed at each time point, and they would not be aware of the training the provider had had. Field data collectors were also blinded to study arm allocation as much as possible, although some might have determined intervention arm during the study.

Some more information about the intervention itself (training), that differed between the two arms, should also be provided, such as how much/number of hours of training in how to deliver a standardized “ABCD” approach for PCC did the providers receive.

Across the three study countries, the PCC training was only given to ANC providers in the intervention arm. Providers in the control arm did not receive any training. A standardized manual and job aids guided the three-day training. Due to limitations in word count, we did not describe this training in detail, but the protocol provides more details on the content and approach used in the training. The training package was made publicly available here following the study.

It would strengthen the study if the researchers explained their choice of ICC=0.20.

The selected sample size accounted for a clustering effect (ICC=0.20) on the clinic level so as not to underestimate the needed sample size. Originally, we accounted for ICC up to 0.20 in the sample size calculations (90 + 90 facilities, each providing 10 patients at each data collection period, power 80%, significance level 5% with assumed levels of 30% and 20%, respectively). In the initial analysis we observed ICC values close to 0 among the provider-level data because many sites had a single provider and there was a relatively large number of small clusters. In terms of the clustering on the client level data when reporting on provider actions, problems with convergence arose when running two-level logistic regressions. This made adjustment for client characteristics less meaningful, however, we include these adjusted results in the models and report AORs. We explain in the manuscript on page 12 that there is only a negligible clustering on facility level but indicate that we do conduct multi-level regression for the client-level outcomes where the clustering effect was higher.

With respect to the outcomes, the researchers state that the instruments were validated “where possible”. In principle, it is always possible to perform validation of an instrument. In this manuscript, it should be stated which instruments, measuring which outcomes, were validated and which were not.

Some components of the questionnaires were based on validated instruments, specifically the self-efficacy questions and the questions on interpersonal communication. In addition, validated constructs of person-centred communication are operationalized into questions to assess the primary study outcome. We have clarified this in the manuscript. The questions on knowledge, attitudes, and practice (KAP) on FGM prevention and care, as well as the health facility checklist have not been validated. Some additional work is underway to develop a validated KAP instrument based on a previous version used in formative research in Guinea.

All outcomes are self-report, either by provider or by client. It is stated that the primary outcome is delivery of the ABCD approach by ANC providers: on page 62-63 we see that it is actually four outcomes (facility preparedness to provide FGM services, provider utilization of WHO handbook of FGM, providing FGM related care (i.e. discourage FGM, look for FGM when examine, record FGM in file), deliver ABCD package (as perceived by clients). I suggest a revision or explanation of this.

To clarify, there are two main study outcomes. The first one is a composite measure related to the delivery of FGM prevention and care services, with the prevention aspect operationalized into the ABCD approach. The second primary outcome relates to health facility preparedness as it relates to the Level One intervention. This has been clarified in the manuscript.

There are a number of secondary outcomes. The results and tables should be ordered with the main outcome presented first, followed by the secondary outcomes. And the discussion should highlight the result for the main outcome.

We have reordered the table of results so that the primary outcomes are presented first, and we clarify which are the primary, secondary and other outcomes. The text describing the objectives and results follows this same ordering. We thank the reviewer for this comment and hope it is clearer now.

It is noteworthy that the intervention did not succeed in improving providers' FGM-related knowledge, self-efficacy, FGM-related attitudes (table3), and these results deserve more discussion.

We believe that the main reason for the intervention not improving providers' self-efficacy and FGM-related attitudes was because these were already high at baseline. In terms of knowledge, these were low throughout the study and upon further investigation of the responses, the questions with visual depictions of typology appeared to be consistently incorrect, indicating challenges in differentiating FGM types based on drawn images displayed on a tablet or low recall on this aspect of knowledge. The discussion section provides additional discussion on these points.

Concerning reporting, it would be useful if there was clearer information about any use of restrictions in connection with the randomization procedure, mechanisms used to implement the random allocation sequence and steps taken to conceal the allocation, who were involved in the allocation sequence (generated allocation sequence, enrolled clinics, assigned clinics).

The randomization was based on a list of clinics sorted by client load based on data from the Ministry of Health in each country. The facilities were paired and then randomized by the study statistician using the uniform random number function in STATA 17. The lists of intervention and control sites were shared with the Principal Investigators in each country to enable them to implement the Level Two intervention in the intervention arms and organize data collection.

Finally, there appears to have been 3 funders (government of Norway, government of UK, WHO/HRP). One funder, WHO/HRP, appears to have been heavily involved in the study, including design, implementation, protocol, data management and analysis, interpretation and manuscript. The potential for bias is considerable and must be addressed.

We added a statement to the manuscript explaining the following:
WHO/HRP played a coordinating role for successful implementation of this research study. The study protocol was developed in collaboration with country research teams. The data collection platform was developed and maintained by an outsourced vendor (FirstData, LLC, Kenya); data management was coordinated by the local implementing partners (CERREGUI, DARS and University of Nairobi) and statistical data analysis was conducted by an external statistician (Dr. Max Petzold, Gothenburg University). All these functions were conducted with utmost integrity following ICH-GCP guidelines.

Reviewer: 2
Prof. Alex McConnachie, University of Glasgow

Comments to the Author:
Balde et al present the results of a cluster randomised trial of training in person-centred communication in relation to the prevention of female genital mutilation in three countries in Africa. This review considers the use of statistics in the paper.

Generally, these are quite good, though I think a few modifications could be made that would improve the paper.

The authors report compliance with the CONSORT guidelines, and provide a checklist in support of this. However, they have used the regular version, not the checklist of cluster randomised trials, which would be more appropriate in this case.

Thank you for this comment. We have updated the CONSORT checklist included with the revised manuscript.

There is a section about blinding, though for this sort of trial, I feel that it is important, if possible, for the outcome assessors to be blind to the allocation of the clinic. Given that only intervention clinics were assessed at three months, it may be that this was not possible. Either way, I think it is important to be clear on this point.

Attempts were made to blind clinic managers, ANC providers and their clients to study arm allocation. Since both study arms received the level one intervention component at baseline, and the providers and managers at control sites were unaware of the training that took place at intervention sites, it is conceivable that they were not aware of their study arm. Presumably, intervention clients would assume they were the intervention arm, but they were also not aware of what might have been offered to other sites. ANC clients, however, were completely blinded as to study arm allocation since a distinct set of clients was interviewed at each time point and they would not be aware of the training the provider had had. Field data collectors were also blinded to study arm allocation as much as possible, although some might have determined intervention arm during the study.

The study was designed with a pre-specified sample size, which accounted for a healthy dose of clinic-level clustering, which is good. However, I believe a little more detail is required; there is currently not enough information to replicate the calculation (the protocol is the same). In particular, while the authors state that the study was design to have power to detect a 10% difference, the

sample size required will depend in the outcome rates in the two groups, not just on the difference. Could the assumed percentages for both groups be made clear?

The difference between groups was assumed to be 10%, and we thank the reviewer for noting that we had not included the proportions in both groups. In fact, we had used 20% and 30% for the calculation, and we have added this information in the manuscript.

The analyses presented do not take account of clustering, on the basis that any clustering was negligible, and the results from multilevel models were “almost equal”. That sounds reasonable, but I suggest that for the primary analysis paper, it would be better to report the results from multilevel models, or otherwise take account of clustering. If the results are almost equal, it will make no difference to the conclusions. And, it will allow reporting of the ICCs observed in this study, which is one of the recommendations of the CONSORT extension for cluster RCTs. At a minimum, these results ought to be provided as part of the supplementary materials.

We thank the reviewer for the comment. The original decision to not conduct multi-level analyses was based on the analysis of clustering in the provider level data. Since many sites only had one provider in the study, the clustering was not meaningful. However, upon further consideration and analysis, we have concluded that the results using client-level data should account for clustering. We have run these analyses and presented the results in Table 4 to account for clustering. In some cases, particularly when clients report on actions by their providers, there were convergence problems during the iterative process when estimating the multilevel models. In these cases, the corresponding ordinary model was chosen. We have now included the ICCs for the various analyses where multilevel models are presented, and we have explained this more clearly in the manuscript.

There is a statement about checking continuous covariates for linearity, but it is not clear how this was done.

We have revised the manuscript to indicate that we used the Box-Tidwell test (boxtid command in Stata) to test for linearity between the outcome and the continuous variables. As these results were not significant, it can be assumed that the assumption is correct.

The term “multiple variable [logistic] regression” is used, though I feel “multivariable” is more commonly used.

The use of “multiple variable” and “multivariable” were used interchangeably but we agree that multivariable is easier to interpret and tends to be the preferred choice in most studies and we have edited the manuscript accordingly.

Under “Characteristics of study sites and participants”, there is a statement that “...characteristics were not different from the 17 ANC clinics that were enrolled at baseline but that subsequently were lost to follow-up.” No evidence is provided in support of this – could a table be added to the supplements to show this, perhaps? There is a similar issue when talking about the ANC providers who did or did not receive PCC training – the differences and similarities between these providers is discussed, but no data are given in support of these statements.

We have included data on these 17 sites in a supplementary file.

In the same section, there is a typo: “(64%, n=14)”; should this be n=149?

Thank you for noting this, we have revised the figure accordingly.

In the section “Health facility preparedness”, and for much of the subsequent sections, several results are presented, with p-values, though it should be made clear in which table(s) these data are presented.

We have revised Table 4 to organize results by outcome level and hope these changes clarify.

There is a comment that scores for FGM-related knowledge ranged from 1.6 to 2.5, but it is also stated that the mean score in the intervention arm was 2.5, but this would not be consistent with the upper end of the range being 2.5. Perhaps I am misinterpreting something here, so maybe it needs to be clearer.

We have revised the manuscript to clarify that the mean score calculated remains 2.5 and that the FGM-related knowledge-related scores were generally low. We also present percentages of those having correct responses in each group with significance testing for this result.

In Table 3, I think it would be easier to read if the intervention and control arm data were given in separate columns. It is not clear why some of the odds ratios are not reported, and, for the continuous outcomes, why there are no estimates of the mean differences, with 95% CIs and p-values. These should be given for all outcomes.

We assumed this comment related to Table 4. Odds ratios are now presented for most results. We also present results of t-tests with p-values and 95% CIs for the comparison of scale scores of the continuous variables.

Note, given my earlier comments, I feel these should be the intervention effect estimates derived from models allowing for clustering, and the ICCs should be reported – this is important information for others wishing to design trials in the same setting.

We agree and we have added this information as a column in the table.

I feel that Table 3, and the paper as a whole, could make it much clearer what the primary outcome of the study was. In the methods, it is stated that the primary outcome was the delivery of the “ABCD” approach by ANC providers. Since this was as defined in the protocol, it cannot be changed, but it seems an odd choice, since only those in the intervention arm were trained in using this approach. On that basis, it hardly seems like a fair outcome on which to compare the two groups.

While exposure to the training would be expected to increase knowledge and skills, the decision to use delivery of the ABCD approach as the primary outcome and to assess the outcome more than three months after the training was to determine if trained providers would actually apply the approach during routine care several months after being exposed to it. We would not expect providers in control sites to implement the counseling, as the reviewer notes, but we also felt that we needed to establish whether the providers who were trained would implement the counseling as a critical step in understanding the real-world applicability, feasibility and impact of this novel approach. To respond to the second point, we have edited the table so the results are presented in an order that matches the objectives and results to improve clarity.

Reviewer: 1

Competing interests of Reviewer: None

Reviewer: 2

Competing interests of Reviewer: None

VERSION 2 – REVIEW

REVIEWER	McConnachie, Alex University of Glasgow, Robertson Centre for Biostatistics
REVIEW RETURNED	18-Apr-2024

GENERAL COMMENTS	I thank the authors for their consideration of my original comments. I am happy with their responses and have only a few minor comments left. Apologies for not spotting this previously, but I feel the abstract should make a clear statement in relation to the primary outcome of the study. It is good that the authors are using the cluster RCT version of the CONSORT checklist, however, reference 22 still refers to the main CONSORT guideline. It is also good that the authors have added ICC values for relevant outcomes. I note the values are very high (all >0.5). Should this be discussed at all? Is this a limitation, given that the original assumption was ICC=0.2?
--

VERSION 2 – AUTHOR RESPONSE

Reviewer: 2

Prof. Alex McConnachie, University of Glasgow

Comments to the Author:

I thank the authors for their consideration of my original comments. I am happy with their responses and have only a few minor comments left.

Apologies for not spotting this previously, but I feel the abstract should make a clear statement in relation to the primary outcome of the study.

Thank you for this observation. We have revised the abstract to highlight findings from the study's primary outcome.

It is good that the authors are using the cluster RCT version of the CONSORT checklist, however, reference 22 still refers to the main CONSORT guideline.

We have revised the reference accordingly.

It is also good that the authors have added ICC values for relevant outcomes. I note the values are very high (all >0.5). Should this be discussed at all? Is this a limitation, given that the original assumption was ICC=0.2?

Thank you for this observation. We have revised the limitations section to indicate that while the sample size was calculated based on an ICC of 0.20, the observed ICCs were all above 0.50 leading to statistically conservative conclusions.

Reviewer: 2

Competing interests of Reviewer: None